# Newborns with Favourable Outcomes after Perinatal Asphyxia Have Upregulated Glucose Metabolism-Related Proteins in Plasma

**DOI:** 10.3390/biom13101471

**Published:** 2023-09-30

**Authors:** Ping K. Yip, Michael Bremang, Ian Pike, Vennila Ponnusamy, Adina T. Michael-Titus, Divyen K. Shah

**Affiliations:** 1Centre for Neuroscience, Surgery and Trauma, Blizard Institute, Barts and The London School of Medicine and Dentistry, Queen Mary University of London, London E1 2AT, UK; vennilaponnusamy@nhs.net (V.P.); a.t.michael-titus@qmul.ac.uk (A.T.M.-T.); d.shah@qmul.ac.uk (D.K.S.); 2Proteome Sciences PLC, Coveham House, Surrey KT11 3EP, UKian.pike@proteomics.com (I.P.); 3St. Peter’s Hospital (Ashford and St. Peter’s Hospitals NHS Foundation Trust), Chertsey KT16 0PZ, UK; 4Neonatal Unit, Royal London Hospital, Barts Health NHS Trust, London E1 1FR, UK

**Keywords:** hypoxic-ischaemic encephalopathy, therapeutic hypothermia, plasma, tandem mass spectrometry, protein, glucose metabolism

## Abstract

Hypoxic-ischaemic encephalopathy (HIE) is an important cause of morbidity and mortality globally. Although mild therapeutic hypothermia (TH) may improve outcomes in selected babies, the mechanism of action is not fully understood. A proteomics discovery study was carried out to analyse proteins in the plasma of newborns with HIE. Proteomic analysis of plasma from 22 newborns with moderate-severe HIE that had initially undergone TH, and relative controls including 10 newborns with mild HIE who did not warrant TH and also cord blood from 10 normal births (non-HIE) were carried out using the isobaric Tandem Mass Tag (TMT^®^) 10plex^TM^ labelling with tandem mass spectrometry. A total of 7818 unique peptides were identified in all TMT10plex^TM^ samples, translating to 3457 peptides representing 405 proteins, after applying stringent filter criteria. Apart from the unique protein signature from normal cord blood, unsupervised analysis revealed several significantly regulated proteins in the TH-treated moderate-severe HIE group. GO annotation and functional clustering revealed various proteins associated with glucose metabolism: the enzymes fructose-bisphosphate aldolase A, glyceraldehyde-3-phosphate dehydrogenase, phosphoglycerate mutase 1, phosphoglycerate kinase 1, and pyruvate kinase PKM were upregulated in newborns with favourable (sHIE+) outcomes compared to newborns with unfavourable (sHIE−) outcomes. Those with favourable outcomes had normal MR imaging or mild abnormalities not predictive of adverse outcomes. However, in comparison to mild HIE and the sHIE− groups, the sHIE+ group had the additional glucose metabolism-related enzymes upregulated, including triosephosphate isomerase, α-enolase, 6-phosphogluconate dehydrogenase, transaldolase, and mitochondrial glutathione reductase. In conclusion, our plasma proteomic study demonstrates that TH-treated newborns with favourable outcomes have an upregulation in glucose metabolism. These findings may open new avenues for more effective neuroprotective therapy.

## 1. Introduction

Hypoxic-ischaemic encephalopathy (HIE) remains a devastating problem in term infants, that causes significant morbidity and mortality in newborns after perinatal asphyxia. It has been estimated that up to 3.8 per 1000 live births are affected in industrialised nations [1] and those newborns that survive often suffer from cerebral palsy, cognitive impairments, and epilepsy. Currently, the only available treatment is therapeutic hypothermia (TH), with numbers needed to treat of 9 for reduced death and severe disability [2]. Therefore, there is a need to further understand the pathology of the disease processes and/or the mechanisms of action of TH at the molecular level, so that more infants can be successfully treated.

Plasma has been successfully used to study newborns with HIE, and predetermined protein targets are studied using the technique of immunoassay [3,4,5,6]. Studies have frequently focused on markers related to neurones [7,8], astrocytes [8,9], and inflammatory markers [8,10] using a single-plex enzyme-linked immunosorbent assay (ELISA). Several independent groups using this technique have identified specific protein markers related to brain injury in the blood samples of newborns with HIE [11]. For example, increased levels of S100β [11], NSE [11], UCH-L1 [12], GFAP [8] and NfL [13,14] was identified. However, conventional ELISA can only test a single known analyte at a time due to analyte-antibody specificity. Although improvements have resulted in multiplex immunoassays testing multiple analytes in a single run, they cannot test multiple patient samples within the single run, giving rise to potential measurement variation between patients [15]. Furthermore, the analytes chosen are often restricted by current knowledge and understanding.

Improvements in mass spectrometry proteomic analysis have enabled more effective detection of low-abundance proteins and an unbiased analysis of samples [16]. The commercially available multiplexed proteomics technique involving the Tandem Mass Tag (TMT) 10plex™ labelled mass spectrometry can avoid all these issues by the ability to analyse complex protein mixtures from up to 10 patients simultaneously, with high resolution, specificity, and reproducibility, to enable the identification of selective protein targets from thousands of proteins in an unbiased format [16,17].

This study using the TMT10plex^TM^ labelled liquid chromatography-tandem mass spectrometry aimed to identify a protein signature that would distinguish newborns at risk of adverse outcomes after perinatal hypoxia-ischemia.

## 2. Materials and Methods

### 2.1. Recruitment

Recruitment of newborns was carried out at four centers in the United Kingdom (The Royal London Hospital, Homerton University Hospital, University Hospital Southampton, and Norfolk and Norwich University Hospital) during the period of January 2014 to January 2016, with Research Ethics Committee approval (Bromley REC ref: 13/LO/17380), as part of the Brain Injury Biomarkers in NewbornS (BIBiNS) study. Blood from newborns with moderate-severe HIE treated with mild therapeutic hypothermia (TH), based on the standard cooling criteria (UK TOBY Cooling Register Clinician’s Handbook) [18], as per local clinical practice was collected. Furthermore, the mild HIE group comprised of newborns admitted to the neonatal unit for treatment with intravenous fluids. These infants had grade 1 encephalopathy and/or had required resuscitation and/or had low cord pH and/or had a metabolic acidosis or raised lactate but did not fulfil standard cooling criteria, hence were not treated with TH (mHIE group). The “normal” group (ncord group) consisted of cord blood from babies born after 36 weeks gestation either vaginally or after elective caesarean section with normal Apgars who required no resuscitation or admission to the neonatal unit. Pregnancies affected by pre-eclampsia, diabetes and congenital malformations or co-morbidities were excluded.

Newborns who had suffered moderate-severe HIE and had undergone TH were subdivided into those who had cerebral magnetic resonance imaging (MRI) predictive of favorable outcomes (sHIE+ group) and those who had cerebral MRI predictive of unfavorable outcomes (sHIE− group). Cerebral MR images were rated independently by a neuroradiologist and a neonatologist with expertise, in a blinded fashion as previously described [19], using a system validated for use in this group of babies [20]. “Cerebral MRIs predictive of an unfavourable outcome had reversed or abnormal signal bilaterally on T1/T2 weighted sequences in the posterior limb of the internal capsule (PLIC), multifocal or widespread abnormal signal intensity in the basal ganglia and thalami (BGT), and/or severe widespread white matter (WM) lesions including infarction, haemorrhage and long T1 and T2 (sHIE− group). Infants with images predictive of favourable outcome had normal images or less severe patterns of injury than those described above (sHIE+ group)” [19,20]. Consensus was reached in cases of disagreement. Furthermore, we have demonstrated in this cohort of newborns that cerebral MRI has been shown to be highly predictive of later neurodevelopmental outcomes [19].

### 2.2. Plasma Sample Preparation and TMT10plex™ Labelling

All blood samples were collected in spray-coated K_2_EDTA tubes, centrifuged at 1000× *g* for 10 min, and then the plasma was aliquoted into fresh tubes and stored at −80 °C until further processed. Sixty-eight plasma samples were initially provided to Proteome Sciences Ltd. Plasma samples with substantial haemolysis scores of 4–5 (Figure 1A,B) were excluded for further processing as haemolysis can affect protein expression levels [21]. Depletion of albumin and IgG from each 10 μL plasma sample was carried out using the Pierce™ Top 2 Abundant Protein Depletion Spin Columns (Pierce Biotechnology, Rockford, IL, USA) according to the manufacturer’s instruction, to enable broader proteome profiling. The protein concentration was determined using the Bradford assay according to the manufacturer’s instructions and the samples were visualized on Coomassie stained (Imperial Stain, Pierce) SDS-PAGE 4–20% gradient gels (Criterion, Bio-Rad Laboratories Ltd., Watford, UK) (Figure 1C,D). A final number of 44 samples was selected for LC-MS/MS analysis, with the samples chosen consisted of 12 ncord, 10 mHIE, 12 sHIE+ and 10 sHIE− specimens.

Forty-four albumin and IgG-depleted plasma samples and a reference sample created from a pooling of 37 plasma samples from all patient groups were prepared as previously described [22]. Briefly, plasma samples were reduced with 1 mM tris(2-carboxyethyl)phosphine (TCEP, 1 h in 55 °C), alkylated with 7.5 mM iodoacetamide (1 h at room temperature (RTP)), and digested with trypsin (1:25 weight ratio, approx. 18 h in 37 °C) to generate peptides. Peptides from each sample were labelled with one of a 15 mM TMT10plex™ set of isobaric mass tags (1 h at RTP), then the reaction was quenched with 0.25% hydroxylamine (15 min at RTP). The samples were desalted and purified using RP18 columns and SCX purification, respectively.

### 2.3. Liquid Chromatography Tandem Mass Spectrometry (LC-MS/MS)

Quantitative analysis using LC-MS/MS was carried out as described previously [22]. Briefly, samples were randomly distributed over TMT10plex™ experiments regarding proteomic category, haemolysis rating, and albumin rating. Each TMT10plex™ experiment (*n* = 5) was separated into 8 fractions by HPLC-assisted high pH reverse phase chromatography (EC 250/4.6 NUCLEODUR C18 Gravity (Macherey-Nagel) and HPLC system (Waters Alliance 2695)). All fractions were lyophilised to completion. In each TMT-MS3 experiment, an aliquot of reference sample was treated as the 10th sample. Individual samples were divided into a labelling design in which 9 experimental samples were combined with a single reference. All 10-labelled digests were then mixed in equal amounts to form a TMT10plex™ experiment sample. Fractions were analysed in duplicate by LC-MS/MS using the EASY-nLC 1000 system (Thermo Scientific, Loughborough, UK) coupled to an Orbitrap Velos Pro Mass Spectrometer (Thermo Scientific, Loughborough, UK). Re-suspended peptides were loaded onto a nanoViper C18 Acclaim PepMap 100 pre-column (Thermo Scientific, Loughborough, UK) and resolved using an increasing gradient of 0.1% formic acid in acetonitrile through a 50 cm PepMap RSLC analytical column (Thermo Scientific, Loughborough, UK), at a flow rate of 250 nl/min. Peptide mass spectra were acquired throughout the entire chromatographic run (180 min), using the TOP 10 collision-induced dissociation (CID) method, enabling higher collision-induced dissociation (HCD) MS3 scans for accurate quantitation of TMT reporter ions.

### 2.4. Proteomics Data Analysis

Raw data files were submitted to Proteome Discoverer v1.4 (Thermo Scientific, Loughborough, UK) using the Spectrum Files and Spectrum selector node. The SEQUEST HT node was suitably set up to search data against the human UniProtKB database, allowing for variable modification of methionine oxidation. The reporter ions quantifier node was set up to measure the raw intensity values for TMT10plex™ monoisotopic ions (126, 127N, 127C, 128N, 128C, 129N, 129C, 130N, 130C, 131). All raw intensity values were exported for later processing and filtering using in-house software which included options for filtering, normalization, exploratory analysis, biostatistics, and functional analysis. Individual peptide spectrum matches (PSM) were filtered for isolation interference values <50% and TMT^®^ intensity values were median-scaled. Ratios of reporter ion intensities were calculated for each sample relative to the reference sample within each plex and log2-transformed. PSMs belonging to identical peptide sequences were then summarised (median). Subsequently, peptides with ≥35% missing values per patient were removed, and the remaining missing values were replaced using an iterative principal component analysis (PCA)-based approach [23,24]. A Linear Models for Microarray data (LIMMA)-based approach was used to correct batch effects, due to the different TMT^®^ experiments, and the data matrix was median-normalised. Protein values were determined as the trimmed mean of unique, unmodified peptides. Functional analysis was performed using in-house software, calculating the significance of enrichment for GOBP terms.

### 2.5. Western Blot Analysis

To validate the mass spectrometry data, the western blotting technique was carried out on a selected number of protein targets as previously described [25,26]. Fifteen μg of total plasma protein samples, determined with the bicinchoninic acid assay, diluted 1 in 25 with ice-cold lysis buffer (containing 20 mM HEPES, 100 nM NaCl, 100 mM NaF, 1 mM Na_3_VO_4_, 5 mM EDTA, 1% Nonidet P-40 substitute, and 1x protease inhibitor cocktail (Roche cOmplete^TM^)) were loaded onto 12% sodium dodecyl-sulfate polyacrylamide gel then subjected to electrophoresis. Each blot contained mHIE (n = 4), sHIE+ (n = 5–6), and sHIE− (n = 4–5). Furthermore, 1 of 8 (12.5%), 6 of 11 (54.5%), and 5 of 9 (55.6%) of the samples used in the western blotting were not used in the proteomic study to carry out in- and out-of- sample validation for mild, sHIE+ and sHIE− groups, respectively. Thereafter, the protein bands in the gel were transferred onto Hybond™ P nitrocellulose membrane using wet western blot transfer. The quality and quantity of protein bands on the blots were determined using 0.1% Ponceau in 1% (*v*/*v*) acetic acid solution and the images were captured using the Bio-Rad ChemiDoc™ gel system. After washing with PBS-Tween 20 and blocking for 30 min in 5% skimmed milk, incubation of the blot with the antibodies was carried out overnight at room temperature (Table 1). Next day, blots were washed with phosphate-buffered saline (PBS), then incubated with goat anti-mouse and -rabbit horseradish peroxidase for 1 h at RTP (Table 1).

The blot was washed with PBS twice and then once with distilled water, before incubation with ECL™ Prime Western blotting detection reagent (RPN2232) for 5 min and imaged using the Bio-Rad ChemiDoc™ gel system.

Western blots were analysed using the ImageJ software to determine the protein of interest based on specific antibody expression and then normalised against the total protein loaded per lane, determined from the Ponceau S staining. It has been shown for human biosamples that total protein is an effective loading control compared to using a specific endogenous protein control related to a housekeeping gene [27].

### 2.6. Statistical Analysis

Statistical analysis for clinical data was carried out using SPSS (version 28). Continuous variables between groups were compared using ANOVA and proportions were compared using Pearson’s chi-square statistic. Statistical analysis for western blot was carried out in the GraphPad Prism v8 software with either one-way ANOVA followed by post hoc Tukey test or Kruskal-Wallis one-way ANOVA with post hoc Dunn’s test if the data was parametric or non-parametric, respectively. A *p* value of <0.05 was set as significance.

## 3. Results

### 3.1. Plasma Sample Quality

Sixty-eight plasma samples submitted for analysis were initially visually assessed for haemolysis based on a 1–5 scale (5 highest) (Figure 1A,B). The average protein concentration (±SD) of the 68 plasma samples was 50.3 ± 6.8 mg/mL, with a range of 36.3–68.5 mg/mL. Three samples were slightly outside of the range mean ± two-fold SD. The majority of the plasma samples showed a homogenous band pattern that is typical for plasma samples (Figure 1C). The albumin and IgG were successfully depleted in all but six samples, where a significant band at the molecular weight of albumin and IgG was present (Figure 1D). A final number of 44 samples was selected for LC-MS/MS analysis, with low haemolysis based on haemolytic appearance, patient group, protein quantity and sample volume, with an average protein concentration of 48.9 ± 5.7 mg/mL.

### 3.2. Demographic Characteristics

The perinatal characteristics of the newborns used in the study are described in Table 2. As would be anticipated in comparison to the controls or mHIE groups, the newborns with moderate—severe HIE (containing sHIE + and sHIE−) group had a significantly lower Apgar score at 10 min, increased need for need for chest compressions during resuscitation, worse base deficit in the first hour, higher grade of encephalopathy, more inotropic support during the neonatal period, and increased seizure burden. In comparison to the sHIE+, more infants in the sHIE− group had meconium aspiration, seizures, and required inotropic support. TH was commenced within 6 h after birth and time of commencement was not significantly different between the two groups (median age 2.5 h (sHIE+) and 2 h (SHIE−)).

Of the 12 babies in the sHIE+ group, 6 had normal cerebral MRI scans, three had mild white matter (WM) abnormalities, one had moderate WM abnormality, one had combined mild basal ganglia and thalami (BGT) and mild WM abnormalities, and one had isolated mild BGT abnormalities. In the sHIE− group, 7 newborns had global brain injury (a combination of moderate to severe posterior limb of the internal capsule (PLIC), BGT and WM abnormalities), two had predominantly BGT involvement and one had predominantly WM involvement. Of these 10 sHIE− newborns, six children developed cerebral palsy, one child had severe communication difficulties with normal gross motor function, one had normal outcome, and follow up outcomes were not available for two. Cerebral MR imaging and later outcomes for the mHIE and the ncord groups were not available.

### 3.3. Cord Blood from Normal Pregnancies Exhibits a Different Protein Profile to the Blood of HIE Newborns

From the 44 samples analysed, 3456 peptides were detected after applying stringent filter criteria, which represented 401 proteins. PCA and heatmap plots using a protein matrix defined by regulated proteins with *p* value < 0.05, exhibited a very clear separation of protein expression in ncord plasma in comparison to the HIE groups, that all had overlapping protein profiles (Figure 2A,B). On comparing the mHIE to ncord samples, there were 2 down-regulated and 2 up-regulated proteins (Table 3). The top 2 most down-regulated proteins (UniProt ID, log2 fold change) in the mHIE group were hepatocyte growth factor-like protein (P26927, 0.71 FC) and cholinesterase (P06276, 0.76 FC). In contrast, the top 2 most up-regulated proteins (UniProt ID) affected by mild HIE were isoform 2 of glutathione S-transferase omega-1 (P78417-2, 1.58 FC) and transaldolase (P37837, 1.87 FC).

On comparing the sHIE samples (n = 22) with the ncord samples (n = 12), only up-regulated proteins were detected (Table 4 and Table 5). The top 2 most up-regulated proteins (UniProt ID, log2 fold change) affected in the sHIE+ group were profilin-1 (P07737, 4.08 FC) and moesin (P26038, 3.67 FC) (Table 4). Meanwhile, the top 2 most up-regulated proteins (UniProt ID, log2 fold change) in the sHIE− outcome were heat shock cognate 71 kDa protein (P11142, 2.41 FC) and insulin-like growth factor-binding protein-2 (P18065, 2.04 FC) (Table 5).

### 3.4. mHIE Plasma Exhibit a Different Protein Profile to Plasma from sHIE+ and sHIE− Groups

Analysis of the HIE groups using the PCA plot showed there was no distinct cluster separation between the mHIE, sHIE+ and sHIE− groups (Figure 3A). The mHIE group exhibited greater cluster separation from the sHIE+ group compared to the sHIE− group, based on the PCA plot (Figure 3A). The loading plot and heatmap confirmed that the mHIE group was in the opposite pole to sHIE+ group, while the sHIE− group was more skewed towards the mHIE group (Figure 3B,C).

Comparison between sHIE+ and mHIE groups revealed 61 proteins that were significantly regulated, with 8 proteins down-regulated and 53 proteins up-regulated (Table 6 and Figure 4A). The top 2 most down-regulated proteins (UniProt ID, log2 fold change) were immunoglobulin heavy constant gamma 4 (P01861, −1.46 FC) and haptoglobin (P00738, −1.44 FC). The top 2 most up-regulated proteins (UniProt ID, log2 fold change) were haemoglobin subunit delta (P02042, 1.91 FC) and heat shock 10 kDa protein, mitochondrial (P61604, −1.57 FC).

Comparing sHIE− and mHIE revealed 10 proteins that were significantly regulated, with 5 proteins down-regulated and 5 proteins up-regulated (Table 7 and Figure 4B). The top 2 most down-regulated proteins (UniProt ID, log2 fold change) affected by sHIE− outcome were immunoglobulin heavy constant gamma 4 (P01861, −1.65 FC) and immunoglobulin heavy constant gamma 3 (P01860, −1.05 FC). In comparison, the top 2 most up-regulated proteins (UniProt ID, log2 fold change) affected by sHIE− outcome were hemoglobulin subunit delta (P02042, 1.42 FC) and mitochondrial heat shock protein 10 kDa (P61604, 1.21 FC).

### 3.5. Comparison between sHIE+ and sHIE− Groups

Comparison within the moderate-severe HIE group who were treated with hypothermia (sHIE+ and sHIE− groups) revealed 31 proteins that were significantly regulated, with 1 protein down-regulated and 30 proteins up-regulated (Table 8 and Figure 4C). The only down-regulated protein (UniProt ID, log2 fold change) affected by sHIE+ outcome was bone marrow proteoglycan (P13727, −0.72 FC). The top 2 most up-regulated proteins (UniProt ID, log2 fold change) linked to sHIE+ outcomes were creatine kinase M-type (P06732, 2.11 FC) and histone H3.1 (P68431, 1.45 FC). Six of these 31 proteins (19%) are related to the glucose metabolism pathway.

### 3.6. Validation of Selected Protein Targets with Western Blotting

Western blot was carried out to validate some randomly selected targets identified in several group comparisons observed from the proteomic analysis, to validate the result in newborn samples. The proteins glyceraldehyde-3-phosphate dehydrogenase (GAPDH), lactotransferrin, β-actin, and phosphoglycerate kinase 1 (PGK1) were all significantly increased in the plasma of sHIE+ infants compared with sHIE− outcomes (Table 8 and Figure 5A–D). Similarly, the following proteins, GAPDH and namely upper band of lactotransferrin were also significantly upregulated in the sHIE+ group compared to the mHIE group (Figure 5A,B). The total protein values (means +/− SD) based on the Ponceau staining for the groups were mHIE, 28,828 +/− 8097 AU; sHIE+, 26,878 +/− 6395 AU; sHIE−, 32,494 +/− 8049 AU.

### 3.7. Biological Processes and Pathways

Enrichment gene ontology for cell component and functional analysis were compared between the sHIE+ and mHIE (Figure 6A,B) or sHIE− groups (Figure 6C,D). In the cell component analysis for the sHIE+ compared to mHIE groups, cytosol, nucleus and cytoplasm were the top 3 main cell components, and glycolysis, gluconeogenesis, and glucose metabolism were the top 3 main functional roles. In contrast, the cell component analysis for the sHIE+ compared to sHIE− were cytosol, vesicle and cytoplasm for the top 3 cell components, and interleukin-12 (IL-12) signalling, glycolysis and gluconeogenesis for the top 3 functional roles.

Using the database for annotation, visualization and integrated discovery (DAVID) functional annotation tools, a high number of regulated proteins were post-translationally modified or involved in processes related to phosphorylation or acetylation, with an enrichment score of 13.89 in one annotation cluster (Figure 7A). A second annotation cluster with an enrichment score of 6.56 was on nicotinamide nucleotide and purine (Figure 7A). Furthermore, a high number of regulated proteins were post-translationally modified or involved in processes related to phosphorylation or acetylation (Figure 7B). After analysis of differential expressed proteins using STRING 10 software, most differentially-expressed proteins interacted mostly with each other and are related to catabolic processes or cytoplasm/cytoskeleton (Figure 7C).

A closer examination of the proteins related to glucose metabolism and catabolism processes in sHIE+ compared to the other groups revealed 12 proteins, which were all enzymes (Table 9). Nine of these enzymes (fructose-bisphosphate aldolase A/B/C, triosephosphate isomerase, glyceraldehyde-3-phosphate dehydrogenase, phosphoglycerate kinase 1, phosphoglycerate mutase, α-enolase, pyruvate kinase) were involved in the glycolytic/gluconeogenesis pathway, with two enzymes (transketolase and transaldolase) involved in the pentose phosphate pathway, and one enzyme (glutathione reductase, mitochondrial) involved in the citric acid cycle (Figure 8).

## 4. Discussion

The plasma proteins from newborns with TH-treated moderate-severe HIE, and controls from mild HIE and non-HIE cord blood were analysed in a proteomic study using TMT10plex™ mass spectrometry. A number of changes in specific proteins were identified and some were positively verified with western blotting. Moreover, protein signatures from the cord blood from normal pregnancies was distinctly different from any of the HIE groups. These data indicate that there is a radical difference in plasma protein profile between the ncord plasma and the samples from newborns with HIE. Within the HIE groups, the protein signature detected in the mHIE group was more distinct from the sHIE+ group than from the sHIE− group.

Significance of enrichment analysis, focused on Gene Ontology Cell Component terms showed that many of the proteins detected were related to the cell component of the cytosol/cytoplasm and nucleus, and that phosphorylation or acetylation or cytoplasm were the top three key categories involved. Furthermore, functional analysis revealed that many of the proteins detected were related to glucose metabolic pathways. In particular, five glucose metabolism-related enzymes were identified to be upregulated in plasma of newborns in the sHIE+ group, compared to the sHIE− group. A further 5 glucose metabolism-related enzymes were upregulated in the sHIE+ group and not the sHIE− group, in comparison to mild HIE. These data suggest that there are no major differences—apart from a cluster of glucose-related proteins—between the plasma profiles of the moderate-severe HIE babies, when separating those with favourable and those with unfavourable outcomes.

### 4.1. Proteomic Study in HIE Newborns

To date there has been only one small published study of blood samples from newborns with HIE (n = 4). Using plasma samples taken prior to TH and comparing them to the healthy control newborns, they showed that haptoglobin and S100A8 were upregulated in HIE patients [28]. Interestingly, S100A8 and S100A9 proteins were also detected within our study, confirming that the S100A8/A9 complex has a role in HIE, and has been suggested to be involved in the amplification of neuroinflammation [29], and cerebral ischemia-reperfusion injury [30,31]. Importantly, one major difference between the two studies was the timing of blood collection. Zhu and colleagues collected the blood samples before TH, whilst in this study, all our TH-treated HIE blood samples were collected after the initiation of TH. We speculate that the additional protein changes observed in this study, particularly the protein changes within the glucose metabolism, would most likely be influenced by the progression of the pathophysiology and/or TH treatment.

### 4.2. Glucose Metabolism

Within this proteomic study, the expression of proteins associated with glucose metabolism pathways was consistently upregulated. Although this is the first time a significant change in proteins related to the glucose metabolism pathway has been identified in babies with HIE, it is not surprising, since glucose along with oxygen are key substrates for normal brain function. Also, several studies have shown that initial dysglycemia has been detected in infants with neonatal encephalopathy. A few studies have suggested that hyperglycemia associated with HIE may affect brain microstructure and seizures [32,33,34,35]. It is also recognised that hypoglycemia is associated with neonatal brain injury [32,33,36]. Further research in this area is required to fully understand the role of dysglycemia in HIE.

In situations associated with deprivation of glucose and/or oxygen, 6-phosphogluconate dehydrogenase and transaldolase, which are involved in the pentose phosphate pathway (PPP) are upregulated enzymes can undergo ‘anaerobic’ glucose metabolism to generate the largest amount of NADPH in mammalian cells for ATP production [37].

Also, within the glucose/glycogen metabolism pathways, several enzymes including fructose bisphosphate aldolase, triosephosphate isomerase, GAPDH, and phosphoglycerate kinase 1 (PGK1), phosphoglycerate mutase, α-enolase and pyruvate kinase were upregulated. In particularly, GAPDH, PGK1 and pyruvate kinase, which are involved in the production of NADPH or ATP have recently been suggested to have many additional cellular functions apart from the simple glycolysis role such as regulation of gene expression, cell cycle progression and proliferation, apoptosis, T-cell activation and ion channel opening [38,39,40].

### 4.3. Unsuitability of Cord Blood from Normal Pregnancies as Control for This Study

Cord blood resulting from normal pregnancies has been used in various studies as a control group for HIE newborns [12,41,42,43] or as a “time zero” internal control [8,44]. However, as demonstrated in this study, the cord blood from ‘normal’ pregnancies has a distinct protein signature to the HIE group. It is not known whether this difference can be attributed to the difference in pathophysiology or to the source of the blood sample, i.e., umbilical cord. However, cord blood does contain a mixture of stem and progenitor cells [45], which is of sufficient quality to be used as a therapeutic agent [46]. Therefore, it would not be surprising that there are unique differences between umbilical cord blood and peripheral blood. Furthermore, we have also shown using next-generation sequencing that the microRNA expression was also significantly different in the HIE newborn group [47]. Therefore, this further highlights the lack of suitability of cord blood to study HIE when compared with peripherally obtained blood. Solutions to this issue would be to use peripheral blood only from both controls and neonatal HIE [48] or if umbilical cord blood was to be used, then it is important to use the umbilical cord blood for all newborns studied [49,50,51,52].

### 4.4. Certain Protein Biomarkers Were Not Detected in HIE Plasma

In our study, no significant changes in the expression of S100b, NSE, UHC-L1, GFAP and NfL as brain injury protein markers were detected. The reason for the discrepancy is not known, but a potential explanation may be due to the origin of the blood samples and the time point of collection of samples, as mentioned above. For example, Massaro and colleagues showed the greatest elevation in S100b at 0 h, and neuron specific enolase (NSE) at 72 h, in newborns with unfavourable outcome [11]. Also, Douglas-Escobar and colleagues showed increased ubiquitin C-terminal hydrolase L1 (UCH-L1) in serum from the umbilical cord artery at 0–6 h, compared to healthy cord control [12]. Chalak and colleagues showed a significant increase in glial fibrillary acidic protein (GFAP), interleukin (IL)-6 and IL-8 between moderate to severe HIE vs. mild HIE at 6–24 h, during the time of TH, similar to our study [8]. Another potential explanation may be related to the analytic technique, as they used ELISA to detect the protein changes, and there is a discrepancy in the ability to detect proteins between ELISA and proteomic mass spectrometry [53].

### 4.5. Suitability of Certain Endogenous Control Proteins

Interestingly, changes in glyceraldehyde-3-phosphate dehydrogenase (GAPDH) and β-actin expression were identified in this study. These proteins are often used as an internal loading control for western blot analysis. However, their suitability for this use has been recently questioned by other researchers [54]. This is not surprising since similar issues have been identified in other proteomic studies in the changes detected for GAPDH, that are affected by many biological conditions [55]. For example, apart from being involved in energy metabolism, GAPDH displays immunomodulatory roles by suppressing stimulated macrophages [56]. Also, GAPDH has been shown to interact with glutamatergic α-amino-3-hydroxy-5-methyl-4-isoxazolepropionic acid (AMPA) receptors for cortical neurodevelopment, which is still ongoing in newborns [57]. Moreover, β-actin is involved in biological processes. For example, intact beta-actin (~42 KDa) can be converted to cleaved β-actin (32 KDa or 27 KDa) following a very brief focal cerebral ischemia, indicating DNA fragmentation [58]. Moreover, hypothermia can preserve the expression of β-actin mRNA after 10 min of ischemic brain injury in transient bilateral carotid artery occlusion in gerbils, whereas the injury without hypothermia can cause a decrease in β-actin [59]. Therefore, our work suggests that GAPDH and β-actin are not suitable proteins to be used for endogenous controls when studying HIE.

## 5. Limitations and Strengths

As a cross sectional proteomic discovery study, a comparison was made between plasma from four groups of selected newborns; namely ncord, mHIE, sHIE+ and sHIE−. As such, this was not a cohort study. After demonstrating the lack of suitability of the ncord group as controls, comparisons were made between the mHIE group and the sHIE groups. A study such as ours cannot completely disentangle the impact of the severity of HIE from the effect of TH, on the proteomic signatures. Furthermore, because severe HIE occurs in combination with the effects of systemic hypoxia-ischemia, some of the proteomic signatures may not only be related to hypoxia-ischemia of the brain but also to the peripheral tissues. This is because many of the up-regulated enzymes detected are found in all cell types in the body, such as the liver and muscle, so are not specific to the brain. For example, fructose-1,6-(bis)phosphate aldolase A is predominantly located in the muscle and red blood cells and fructose-1,6-(bis)phosphate aldolase B is predominantly located in liver, kidney and small intestine [60]. However, the fructose bisphosphate aldolase C isoform signature, expressed specifically in the brain and neuronal tissue (Table 8), suggesting that brain tissue proteins are also detected [60].

Another limitation of the study is that as sampling could only take place after sensitively obtaining informed consent in this rather unforeseeable clinical condition, a pragmatic approach had to be taken to the timing of blood sampling. This is important given the three pathological phases of HIE, namely; latent phase from minutes to hours; secondary phase from 6 h-3 days, and tertiary phase from weeks to years [61,62]. As the mean (+/−SEM) age of blood sampling was 24 h (+/−21.3) for sHIE−, 18.5 h (+/−11.5) for sHIE+, and 18.7 (+/−7.5) for mHIE groups (Table 2), potentially the range of timing of blood sampling may cover the latent or secondary phase, thus contributing to the results observed.

The strengths of this study are: firstly, the untargeted detection and analysis of hundreds of proteins simultaneously from multiple patients in a single experiment, using TMT10plex^TM^ mass spectrometry, reduces the potential variation between newborn samples; second, the unsupervised analysis of data resulted in the generation of a specific group of proteins, to avoid any bias in protein targets.

## 6. Conclusions

In summary, this is the first proteomic study to analyse plasma samples from newborns with moderate-severe HIE treated with TH. We showed that cord blood from normal pregnancies is an inappropriate control when compared to peripheral blood from newborns with HIE. Also, we provided compelling evidence indicating changes in pathways of glucose metabolism in newborns treated with TH. These findings increase our understanding of why some newborns with moderate-severe HIE treated with TH have favourable outcome, so could give rise to using other therapies in addition to TH, to further improve outcomes for these babies.

## Figures and Tables

**Figure 1 biomolecules-13-01471-f001:**
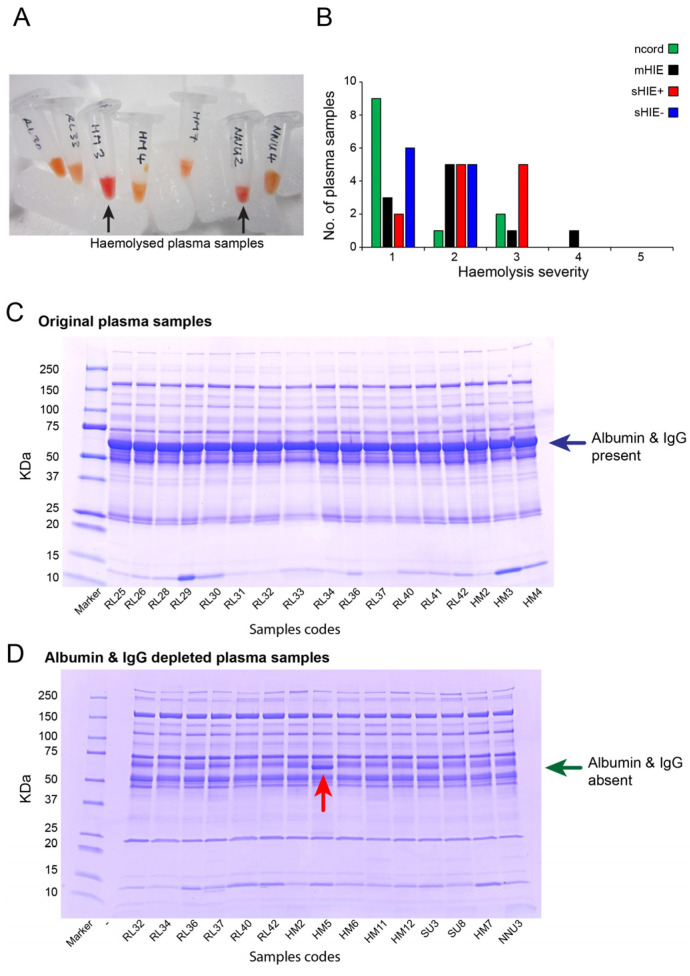
Plasma sample selection and preparation. (**A**) Plasma samples contained various levels of haemolysis (black arrow indicated high haemolysis levels). (**B**) Qualitative analysis of plasma sample haemolysis enables a majority of plasma samples of grade 3 or below to be used. (**C**,**D**) Western blotting stained with Coomassie blue was used to verify albumin and IgG presence in plasma samples ((**C**), blue arrow) and removal using Pierce^TM^ Top 2 Abundant protein depletion spin columns ((**D**), green arrow). One sample (HM5) contained significant amounts of albumin and IgG after purification, so was not included for further analysis ((**D**), red arrow). Ncord, blood from the umbilical cord of non-HIE newborns; mHIE, peripheral blood from newborns with mild HIE and not treated with TH; sHIE+, peripheral blood from moderate-severe HIE newborns treated with TH and with a favourable outcome; sHIE−, moderate-severe HIE newborns treated with TH and with an unfavourable outcome.

**Figure 2 biomolecules-13-01471-f002:**
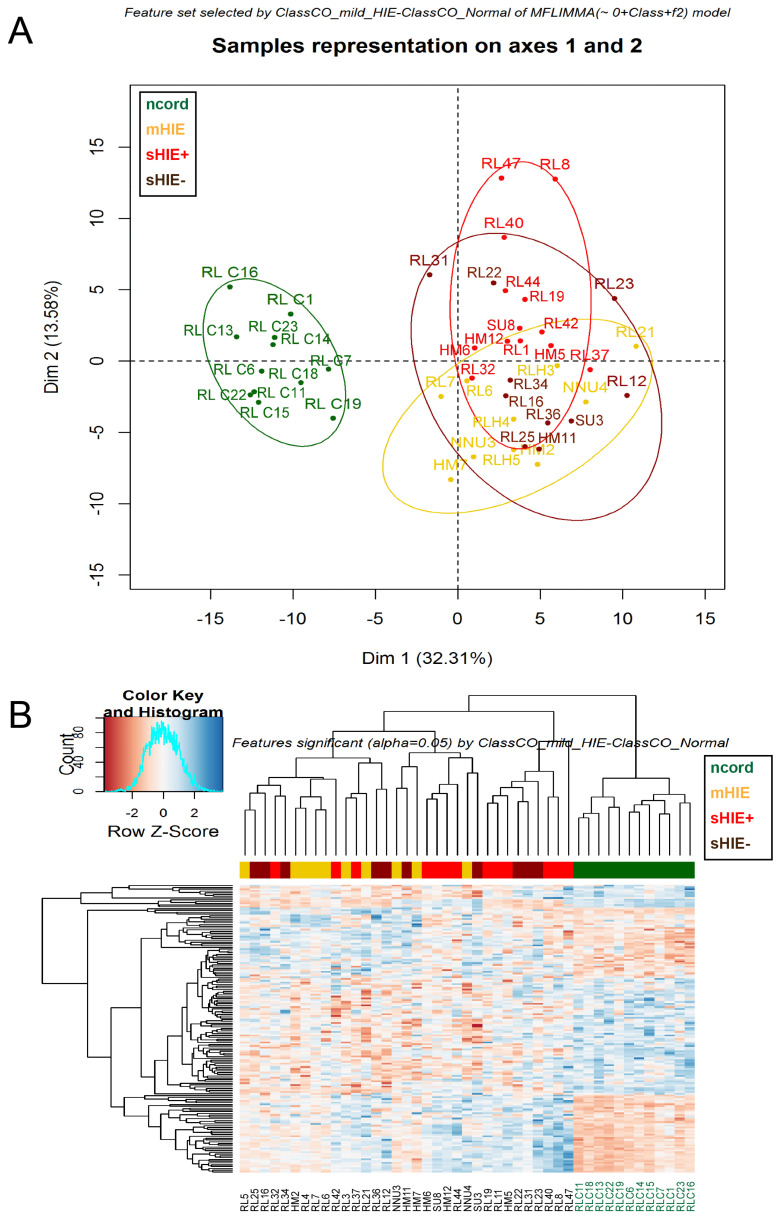
Differences in ncord plasma compared to newborn babies with HIE. (**A**) PCA score plot in first two principal components of protein matrix defined by regulated proteins with *p*-value ≤ α = 0.05. (**B**) Heatmap of the same matrix. Ncord, blood from the umbilical cord of non-HIE newborns; mHIE, peripheral blood from newborns with mild HIE and not treated with TH; sHIE+, peripheral blood from moderate-severe HIE newborns treated with TH and with a favourable outcome; sHIE−, moderate-severe HIE newborns treated with TH and with an unfavourable outcome.

**Figure 3 biomolecules-13-01471-f003:**
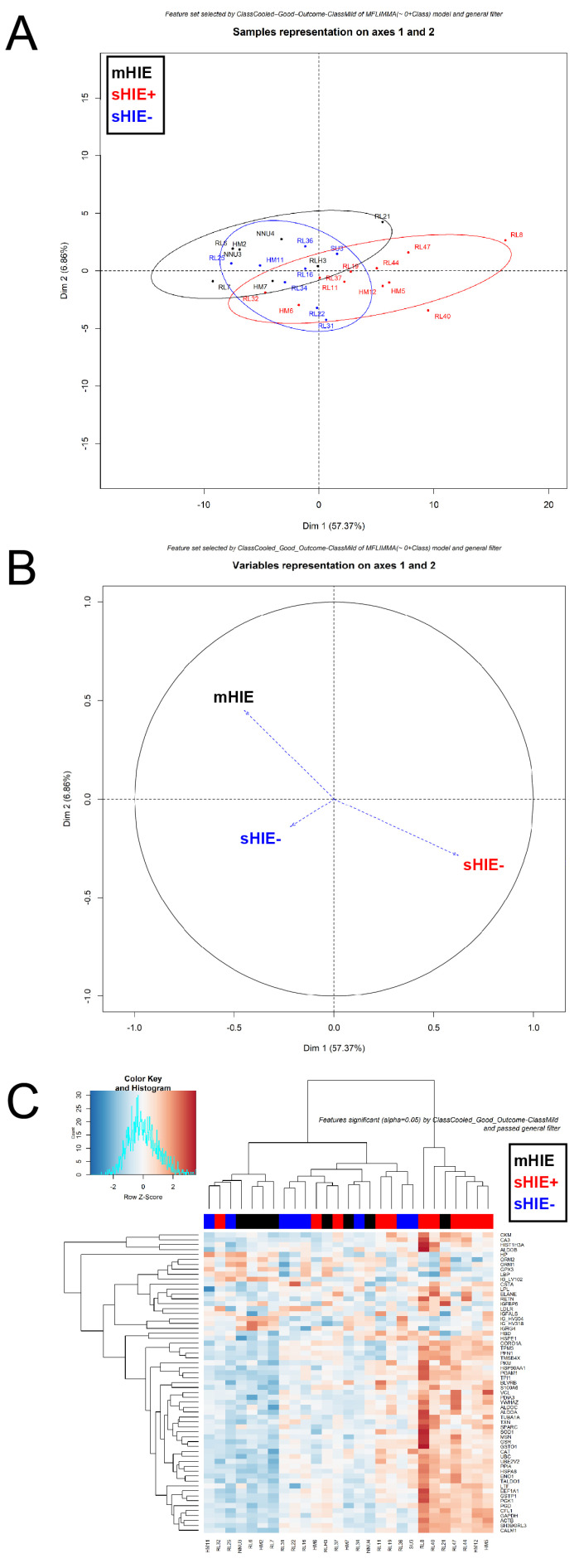
Comparison between plasma proteins of moderate-severe HIE and mHIE. PCA plot (**A**), loading plot (**B**) and heatmap plot (**C**) after normalization. (**A**) PCA plot shows mHIE (black cluster) to be fairly separate from the TH-treated HIE with favourable outcome (sHIE+, red cluster), but TH-treated HIE with unfavourable outcome (sHIE−, blue cluster) overlapped with the mHIE and sHIE+ clusters. (**B**) Loading plot confirmed that the sHIE+ was opposite to the mHIE and that sHIE− was tangent to mHIE and sHIE−. (**C**) Heatmap plot confirmed that sHIE− is fairly separated, with mHIE scattered throughout both cohorts of sHIE. Ncord, blood from the umbilical cord of non-HIE newborns; mHIE, peripheral blood from newborns with mild HIE and not treated with TH; sHIE+, peripheral blood from moderate-severe HIE newborns treated with TH and with a favourable outcome; sHIE−, moderate-severe HIE newborns treated with TH and with an unfavourable outcome.

**Figure 4 biomolecules-13-01471-f004:**
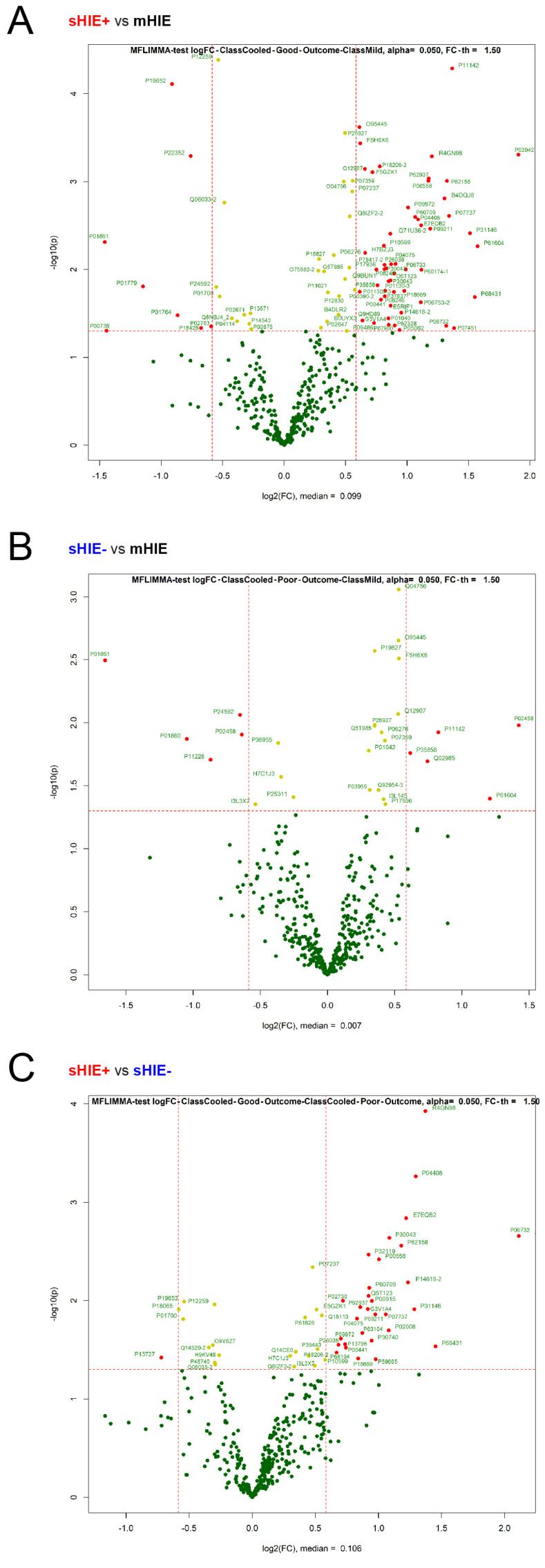
Volcano plot showing regulated features comparing between groups, with estimated log2 fold changes (*x*-axis) versus the log10 *p* values (*y*-axis) for each protein. (**A**) Volcano plot showing significantly regulated features comparing mHIE and sHIE+. (**B**) Volcano plot showing significantly regulated features comparing mHIE and sHIE−. (**C**) Volcano plot showing significantly regulated features comparing sHIE+ and sHIE−. Red and green filled circles indicate proteins that have a significant and non-significant change, respectively. Ncord, blood from the umbilical cord of non-HIE newborns; mHIE, peripheral blood from newborns with mild HIE and not treated with TH; sHIE+, peripheral blood from moderate-severe HIE newborns treated with TH and with a favorable outcome; sHIE−, moderate-severe HIE newborns treated with TH and with an unfavourable outcome.

**Figure 5 biomolecules-13-01471-f005:**
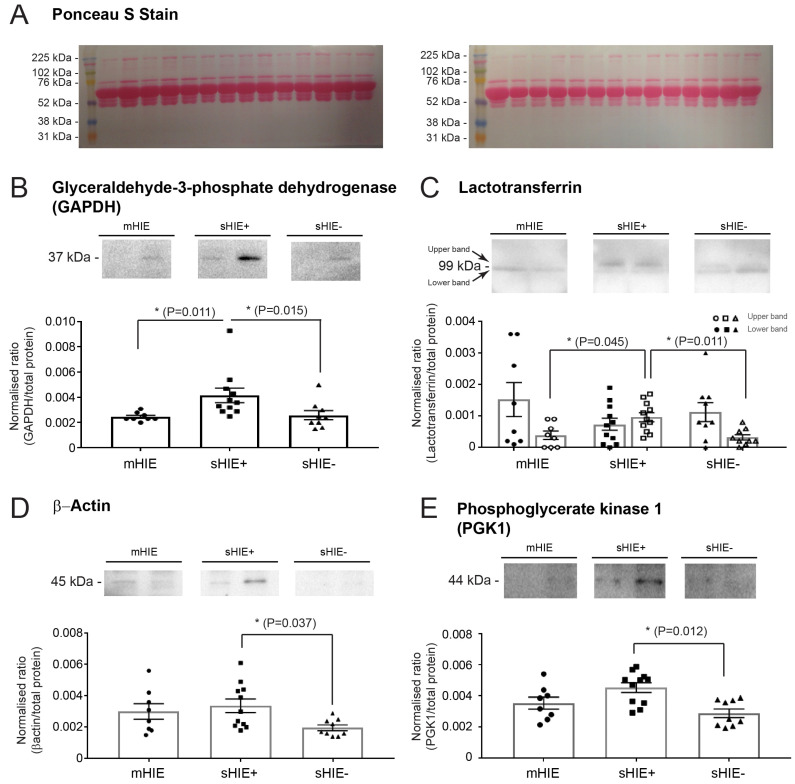
Western blot validation of proteomic data. (**A**–**E**) Western bot analysis of randomly selected proteins between mHIE, and sHIE+ or sHIE−. (**A**) Ponceau S staining of 2 independent blots with each lane randomly allocated to a patient’s plama sample, (**B**) glyceraldehyde-3-phosphate dehydrogenase (GAPDH), (**C**) lactotransferrin, (**D**) β-actin, and (**E**) phosphoglycerate kinase 1. N = 8–11 per group. mHIE, peripheral blood from newborns with mild HIE and are not treated with TH; sHIE+, peripheral blood from moderate-severe HIE newborns treated with TH and have a favourable outcome; sHIE−, moderate-severe HIE newborns treated with TH and have an unfavourable outcome. * = *p* < 0.05. Original Western Bolt Figures can be found in Appendix A.

**Figure 6 biomolecules-13-01471-f006:**
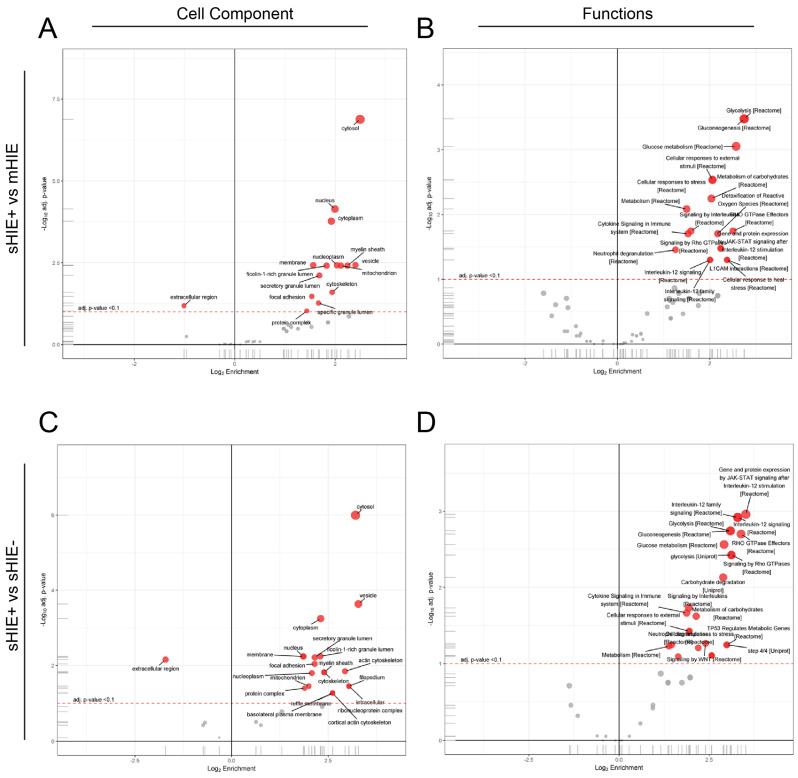
Enrichment Gene Ontology Cell Component and functional analysis volcano plots. (**A**,**B**) Enrichment of Gene Ontology Cell Component (**A**) and functional analysis (**B**) comparing protein levels from sHIE+ with mHIE. (**C**,**D**) Enrichment of Gene Ontology Cell Component (**C**) and functional analysis (**D**) comparing protein levels from sHIE+ with sHIE−. Data points are showing median log2FC against significance of enrichment of protein (-log10adj *p*-value), highlighting the pathways found to be significantly over-represented (shifted to the right side of zero) and under-represented (shifted to the left side of zero). Red filled circles represent significant pathways hits (Adjusted *p* value < 0.05). Grey circles were not significant.

**Figure 7 biomolecules-13-01471-f007:**
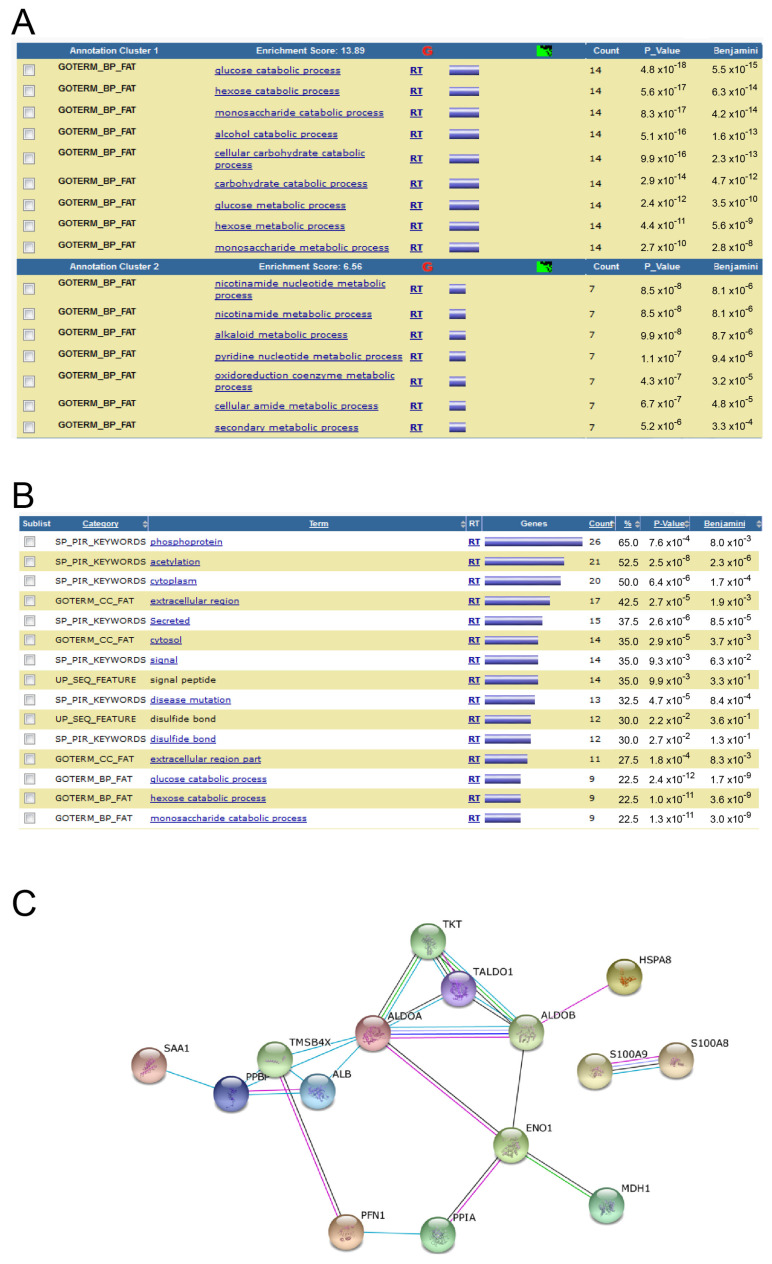
Functional annotation charts and interactomic analysis. (**A**) Clustering of regulated proteins based on the protein matrix using DAVID. Selection criteria: # of summarised peptides ≥ 3. Only two clusters with the best enrichment score out of 35 are shown. (**B**) Regulated proteins based on the protein matrix using DAVID. Selection criteria: # of summarised peptides ≥ 3, sorted to number of protein counts per keyword. Only the 15 hits of 202 with the most number of related proteins are shown. (**C**) Network analysis of proteins differentiated expressed in sHIE group using STRING 10 software. Ncord, blood from the umbilical cord of non-HIE newborns; mHIE, peripheral blood from newborns with mild HIE and not treated with TH; sHIE+, peripheral blood from moderate-severe HIE newborns treated with TH and with a favourable outcome; sHIE−, moderate-severe HIE newborns treated with TH and with an unfavourable outcome.

**Figure 8 biomolecules-13-01471-f008:**
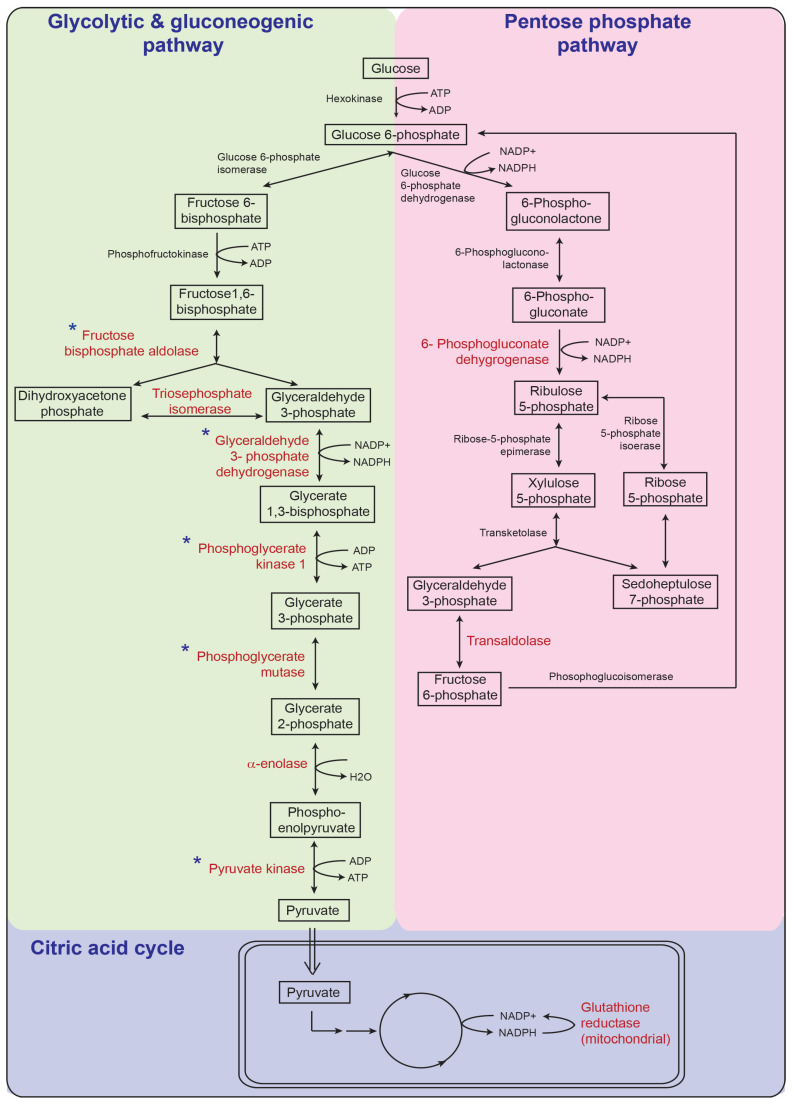
Glucose metabolism pathway affected by the perinatal asphyxia and TH treatment. Glycolysis/gluconeogenesis pathway (green), pentose phosphate pathway (pink) and citric acid cycle (blue). Enzymes (red text) are all proteins significantly up-regulated in sHIE+ newborns compared to non-TH treated controls. Enzymes (blue starred) are upregulated in sHIE+ compared sHIE− group. mHIE, peripheral blood from newborns with mild HIE and not treated with TH; sHIE+, peripheral blood from moderate-severe HIE newborns treated with TH and with a favourable outcome; sHIE−, moderate-severe HIE newborns treated with TH and with an unfavourable outcome.

**Table 1 biomolecules-13-01471-t001:** Primary antibodies used for western blot validation. GAPDH, glyceraldehyde-3-phosphate dehydrogenase, PGK1, phosphoglycerate kinase 1; HRP, Horseradish peroxidase.

Antibody	Species	Dilution	Cat. No	Company
Primary antibodies				
β-Actin	Rabbit	1:250	4970	New England Biolabs, Ipswich, MA, USA
GAPDH	Rabbit	1:500	5174	New England Biolabs, Ipswich, MA, USA
Lactotransferrin	Rabbit	1:500	ab15811	Abcam, Cambridge, UK
PGK1	Rabbit	1:100	ab38007	Abcam, Cambridge, UK
Secondary antibodies				
Mouse immunoglobulins/HRP	Goat	1:10,000	P0447	Dako, Cheshire, UK
Rabbit immunoglobulins/HRP	Goat	1:5000	P0448	Dako, Cheshire, UK

**Table 2 biomolecules-13-01471-t002:** Demography of control newborn babies used in the study, including clinical characteristics of the four groups of babies. Group 1 is blood from the umbilical cord of non-HIE newborns (ncord). Group 2 is peripheral blood from newborns with mild HIE and are not treated with TH (mHIE). Group 3 is moderate-severe HIE babies treated with TH and have a favourable outcome (sHIE+). Group 4 is moderate-severe HIE babies treated with TH and have an unfavourable outcome (sHIE−). Values are expressed as medians (interquartile range), mean ± standard deviation and. %, percentage of the categorical variable. A sentinel event was defined as cord prolapse, cord rupture, uterine rupture, placental abruption or shoulder dystocia as diagnosed by the obstetric team providing clinical care.

	ncord	mHIE	sHIE+	sHIE−	*p*
n	12	10	12	10	
Male	8	4	8	7	0.47
Birth weight (g)	3340 (2790, 3622)	3430 (2886, 3905)	3770 (3172, 4274)	3685 (3472, 3813)	0.39
Apgar at 10 min	10 (10, 10)	9 (9, 10)	4 (3, 6)	4 (3, 5)	<0.001
Post menstrual age (week)	39.1 (38, 39.2)	40 (39, 40.8)	40.6 (39.5, 41.9)	40.5 (40, 41.1)	0.08
Sentinel event	0	1	4	2	0.14
Chest compressions	0	1	5	5	0.016
Cord or pH in first hour	not done	6.97 (6.93, 6.99)	6.93 (6.87, 6.97)	6.88 (6.78, 6.98)	0.33
Cord or base excess in first hour	not done	−12.8 (16.3, −11.4)	−19.1 (−22.8, −16)	−19.5 (−22.0, −18.2)	0.014
HIE grade (0,1,2,3)	12,0,0,0	1,9,0,0	0,0,8,4	0,0,5,5	<0.001
Meconium aspiration (%)	0 (0)	1 (10)	0 (0)	3 (30)	0.05
Positive blood cultures (%)	0 (0)	0 (0)	0 (0)	1 (10)	0.32
Inotropic support (%)	0 (0)	0 (0)	3 (25)	8 (80)	<0.001
Seizuures (%)	0 (0)	0 (0)	7 (58)	9 (90)	<0.001
Age at MRI (d)	not done	not done	8 (7, 10)	8 (8, 9)	0.71
Age at blood sample (hr)	0.5	18.7 ± 7.5	18.5 ± 11.5	24.0 ± 21.3	

**Table 3 biomolecules-13-01471-t003:** List of most reliable plasma proteins between mild HIE and ncord of newborns. Ncord, blood from the umbilical cord of non-HIE newborns; mHIE, peripheral blood from newborns with mild HIE and are not treated with TH.

Protein ID	Gene Name	Protein Name	Number of Peptides	Mild HIE/ncord	
				Log2FC	*p*-Value
P06276	*CHLE*	Cholinesterase	3	0.76	2.44 × 10^−2^
P26927	*HGFL*	Hepatocyte growth factor-like protein	14	0.71	1.26 × 10^−3^
P78417-2	*GSTO1*	Isoform 2 of Glutathione S-transferase omega-1	3	1.58	4.60 × 10^−3^
P37837	*TALDO*	Transaldolase	4	1.87	5.50 × 10^−3^

**Table 4 biomolecules-13-01471-t004:** List of most reliable plasma proteins between sHIE+ and ncord of newborns. Ncord, blood from the umbilical cord of non-HIE newborns; sHIE+, peripheral blood from moderate-severe HIE newborns treated with TH and have a favourable outcome.

Protein ID	Gene Name	Protein Name	Number of Peptides	sHIE+/ncord	
				Log2FC	*p*-Value
P07737	*PROF1*	Profilin-1	4	4.08	2.09 × 10^−8^
P26038	*MOES*	Moesin	3	3.67	8.29 × 10^−13^
P62328	*TYB4*	Thymosin beta-4	3	3.31	2.05 × 10^−6^
Q5T123	*SH3BGRL3*	SH3 domain-binding glutamic acid-rich-like protein	3	3.29	2.00 × 10^−8^
P63104	*YWHAZ*	14-3-3 protein zeta/delta	4	3.09	9.64 × 10^−8^
P04075	*ALDOA*	Fructose-bisphosphate aldolase A	13	2.94	1.21 × 10^−8^
P00558	*PGK1*	Phosphoglycerate kinase 1	7	2.83	3.44 × 10^−8^
P13796	*PLS2*	Plastin-2	5	2.73	9.30 × 10^−8^
P59665	*DEFA1*	Neutrophil defensin 1	3	2.63	1.10 × 10^−4^
E7EQB2	*LTF*	Kaliocin-1 (Fragment)	4	2.45	3.28 × 10^−5^
P04406	*GAPDH*	Glyceraldehyde-3-phosphate dehydrogenase	6	2.42	1.05 × 10^−5^
P60709	*ACTB*	Actin, cytoplasmic 1	5	2.40	1.15 × 10^−6^
P02763	*AGP1*	Alpha-1-acid glycoprotein 1	10	1.52	8.80 × 10^−3^

**Table 5 biomolecules-13-01471-t005:** List of most reliable plasma proteins between sHIE− and ncord of newborns. Ncord, blood from the umbilical cord of non-HIE newborns; sHIE−, moderate-severe HIE newborns treated with TH and have an unfavourable outcome.

Protein ID	Gene Name	Protein Name	Number of Peptides	sHIE−/ncord	
				Log2FC	*p*-Value
P11142	*HSPA8*	Heat shock cognate 71 kDa protein	5	2.41	9.91 × 10^−8^
P18065	*IGFBP2*	Insulin-like growth factor-binding protein 2	7	2.04	6.71 × 10^−6^

**Table 6 biomolecules-13-01471-t006:** List of most reliable plasma proteins between sHIE+ and mHIE newborns. MHIE, peripheral blood from newborns with mild HIE who are not treated with TH; sHIE+, peripheral blood from moderate-severe HIE newborns treated with TH and have a favourable outcome.

Protein ID	Gene Name	Protein Name	Number of Peptides	sHIE+/mHIE	
				Log2FC	*p*-Value
P01861	*IGHG4*	Immunoglobulin heavy constant gamma 4	2	−1.46	4.87 × 10^−3^
P00738	*HP*	Haptoglobin	13	−1.44	4.96 × 10^−2^
P01779	*IG_HV318*	Immunoglobulin heavy variable 3–23	1	−1.15	1.56 × 10^−2^
P19652	*ORM2*	Alpha−1-acid glycoprotein 2	11	−0.91	7.75 × 10^−5^
P01764	*IG_HV304*	Immunoglobulin heavy variable 3–23	2	−0.86	3.32 × 10^−2^
P22352	*GPX3*	Glutathione peroxidase 3	3	−0.76	5.10 × 10^−4^
P18428	*LBP*	Lipopolysaccharide-binding protein	5	−0.67	4.62 × 10^−2^
P02763	*ORM1*	Alpha-1-acid glycoprotein 1	10	−0.59	4.44 × 10^−2^
P00390-2	*GSR*	Glutathione reductase, mitochondrial	1	0.61	1.79 × 10^−2^
O95445	*APOM*	Apolipoprotein M	5	0.61	2.41 × 10^−4^
F5H6 × 6	*GANAB*	Neutral alpha-glucosidase AB	2	0.62	3.67 × 10^−4^
Q9HD89	*RETN*	Resistin	1	0.64	3.81 × 10^−2^
Q12907	*LMAN2*	Vesicular integral-membrane protein VIP36	1	0.66	7.15 × 10^−4^
H7BZJ3	*PDIA3*	Protein disulfide-isomerase	1	0.66	6.45 × 10^−3^
F5GZK1	*EXTL2*	Exostosin-like 2	1	0.72	7.82 × 10^−4^
G3V1A4	*CFL1*	Cofilin-1	2	0.73	4.07 × 10^−2^
P06858	*LPL*	Lipoprotein lipase	1	0.75	9.99 × 10^−3^
P01130-3	*LDLR*	Low-density lipoprotein receptor	3	0.76	1.52 × 10^−2^
P18206-2	*VCL*	Vinculin	5	0.78	6.69 × 10^−4^
P00441	*SOD1*	Superoxide dismutase [Cu-Zn]	2	0.79	2.20 × 10^−2^
P10599	*TXN*	Thioredoxin	1	0.81	5.36 × 10^−3^
P78417-2	*GSTO1*	Glutathione S-transferase omega-1	3	0.82	8.83 × 10^−3^
P26038	*MSN*	Moesin	3	0.82	9.88 × 10^−3^
P08246	*ELANE*	Neutrophil elastase	1	0.82	2.02 × 10^−2^
P04040	*CAT*	Catalase	10	0.82	1.74 × 10^−2^
P68104	*EEF1A1*	Elongation factor 1-alpha 1	1	0.83	9.68 × 10^−3^
P01040	*CSTA*	Cystatin-A	2	0.85	3.60 × 10^−2^
P30043	*BLVRB*	Flavin reductase (NADPH)	2	0.85	1.36 × 10^−2^
P07900	*HSP90AA1*	Heat shock protein HSP 90-alpha	1	0.85	4.26 × 10^−2^
Q71U36-2	*TUBA1A*	Tubulin alpha-1A chain	1	0.86	3.92 × 10^−3^
Q5T123	*SH3BGRL3*	SH3 domain-binding glutamic acid-rich-like protein 3	3	0.87	1.33 × 10^−2^
E5RIF1	*UBE2V2*	Ubiquitin-conjugating enzyme E2 variant 2	1	0.87	2.58 × 10^−2^
F5H265	*UBC*	Polyubiquitin-C	4	0.87	8.71 × 10^−3^
P37837	*TALDO1*	Transaldolase	4	0.89	1.80 × 10^−2^
P06733	*ENO1*	Alpha-enolase	4	0.90	1.11 × 10^−2^
P62328	*TMSB4X*	Thymosin beta-4	3	0.90	4.33 × 10^−2^
P04075	*ALDOA*	Fructose-bisphosphate aldolase A	13	0.91	8.63 × 10^−3^
P05062	*ALDOB*	Fructose-bisphosphate aldolase B	3	0.94	4.88 × 10^−2^
P14618-2	*PKM*	Pyruvate kinase PKM	1	0.95	3.09 × 10^−2^
P18669	*PGAM1*	Phosphoglycerate mutase 1	1	0.98	1.76 × 10^−2^
P63104	*YWHAZ*	14-3-3 protein zeta/delta	4	0.99	9.96 × 10^−3^
P09972	*ALDOC*	Fructose-bisphosphate aldolase C	1	1.01	1.97 × 10^−3^
P60709	*ACTB*	Actin, cytoplasmic 1	5	1.07	2.53 × 10^−3^
P04406	*GAPDH*	Glyceraldehyde-3-phosphate dehydrogenase	6	1.09	2.69 × 10^−3^
P06753-2	*TPM3*	Tropomyosin alpha-3 chain	1	1.11	2.36 × 10^−2^
E7EQB2	*LTF*	Lactotransferrin	4	1.12	3.13 × 10^−3^
P60174-1	*TPI1*	Triosephosphate isomerase	4	1.12	1.01 × 10^−2^
P00558	*PGK1*	Phosphoglycerate kinase 1	7	1.17	9.79 × 10^−4^
P62937	*PPIA*	Peptidyl-prolyl cis-trans isomerase A	6	1.18	9.19 × 10^−4^
P09211	*GSTP1*	Glutathione S-transferase P	1	1.19	3.43 × 10^−3^
R4GN98	*S100A6*	Protein S100	2	1.20	5.15 × 10^−4^
B4DQJ8	*PGD*	6-phosphogluconate dehydrogenase, decarboxylating	1	1.31	1.55 × 10^−3^
P06732	*CKM*	Creatine kinase M-type	9	1.32	4.38 × 10^−2^
P62158	*CALM1*	Calmodulin-1	1	1.33	9.83 × 10^−4^
P07737	*PFN1*	Profilin-1	4	1.34	2.45 × 10^−3^
P11142	*HSPA8*	Heat shock cognate 71 kDa protein	5	1.37	5.16 × 10^−5^
P07451	*CA3*	Carbonic anhydrase 3	3	1.38	4.64 × 10^−2^
P31146	*CORO1A*	Coronin-1A	1	1.51	3.86 × 10^−3^
P68431	*HIST1H3A*	Histone H3.1	1	1.55	2.06 × 10^−2^
P61604	*HSPE1*	10 kDa heat shock protein, mitochondrial	2	1.57	5.44 × 10^−3^
P02042	*HBD*	Hemoglobin subunit delta	2	1.91	4.93 × 10^−4^

**Table 7 biomolecules-13-01471-t007:** List of most reliable plasma proteins between sHIE− and mHIE newborns.mHIE, peripheral blood from newborns with mild HIE and not treated with TH; sHIE−, moderate-severe HIE newborns treated with TH and with an unfavourable outcome.

Protein ID	Gene Name	Protein Name	Number of Peptides	sHIE−/mHIE	
				Log2FC	*p*-Value
P01861	*IGHG4*	Immunoglobulin heavy constant gamma 4	2	−1.65	3.20 × 10^−3^
P01860	*IGHG3*	Immunoglobulin heavy constant gamma 3	5	−1.05	1.35 × 10^−2^
P11226	*MBL2*	Mannose-binding protein C	4	−0.87	1.97 × 10^−2^
P24592	*IGFBP6*	Insulin-like growth factor-binding protein 6	2	−0.65	8.67 × 10^−3^
P02458	*COL2A1*	Collagen alpha-1(II) chain	2	−0.64	1.24 × 10^−2^
P35858	*IGFALS*	Insulin-like growth factor-binding protein complex acid labile subunit	12	0.62	1.74 × 10^−2^
Q02985	*CFHR3*	Complement factor H-related protein 3	1	0.74	2.02 × 10^−2^
P11142	*HSPA8*	Heat shock cognate 71 kDa protein	5	0.82	1.19 × 10^−2^
P61604	*HSPE1*	10 kDa heat shock protein, mitochondrial	2	1.21	4.00 × 10^−2^
P02042	*HBD*	Hemoglobin subunit delta	2	1.42	1.05 × 10^−2^

**Table 8 biomolecules-13-01471-t008:** List of most reliable plasma proteins between sHIE+ and sHIE− newborns. * P62158 is obsolete and it is now P0DP23. sHIE+, peripheral blood from moderate-severe HIE newborns treated with TH and with a favourable outcome; sHIE−, moderate-severe HIE newborns treated with TH and with an unfavourable outcome.

Protein ID	Gene Name	Protein Name	Number of Peptides	sHIE+/sHIE−	
				Log2FC	*p*-Value
P13727	*PRG2*	Bone marrow proteoglycan	2	−0.72	3.80 × 10^−2^
P68104	*EEF1A1*	Elongation factor 1-alpha 1	1	0.67	3.37 × 10^−2^
P26038	*MSN*	Moesin	3	0.68	2.82 × 10^−2^
P09972	*ALDOC*	Fructose-bisphosphate aldolase C	1	0.70	2.43 × 10^−2^
P02730	*SLC4A1*	Band 3 anion transport protein	3	0.72	1.00 × 10^−2^
P13796	*LCP1*	Plastin-2	5	0.74	2.78 × 10^−2^
P00441	*SOD1*	Superoxide dismutase [Cu-Zn]	2	0.74	3.01 × 10^−2^
P04075	*ALDOA*	Fructose-bisphosphate aldolase A	13	0.83	1.52 × 10^−2^
P18669	*PGAM1*	Phosphoglycerate mutase 1	1	0.84	3.87 × 10^−2^
P62937	*PPIA*	Peptidyl-prolyl cis-trans isomerase A	6	0.86	1.17 × 10^−2^
P63104	*YWHAZ*	14-3-3 protein zeta/delta	4	0.87	2.13 × 10^−2^
G3V1A4	*CFL1*	Cofilin-1	2	0.92	1.22 × 10^−2^
Q5T123	*SH3BGRL3*	SH3 domain-binding glutamic acid-rich-like protein 3	3	0.92	8.96 × 10^−3^
P32119	*PRDX2*	Peroxiredoxin-2	9	0.92	3.41 × 10^−3^
P60709	*ACTB*	Actin, cytoplasmic 1	5	0.93	7.41 × 10^−3^
P30740	*SERPINB1*	Leukocyte elastase inhibitor	1	0.95	2.55 × 10^−2^
P00915	*CA1*	Carbonic anhydrase 1	1	0.95	1.01 × 10^−2^
P09211	*GSTP1*	Glutathione S-transferase P	1	0.98	1.38 × 10^−2^
P59665	*DEFA1*	Neutrophil defensin 1	3	0.98	3.95 × 10^−2^
P00558	*PGK1*	Phosphoglycerate kinase 1	7	1.00	3.81 × 10^−3^
P07737	*PFN1*	Profilin-1	4	1.06	1.39 × 10^−2^
P02008	*HBZ*	Hemoglobin subunit zeta	3	1.08	2.00 × 10^−2^
P30043	*BLVRB*	Flavin reductase (NADPH)	2	1.09	2.30 × 10^−3^
P62158 *	*CALM1*	Calmodulin-1	1	1.18	2.76 × 10^−3^
E7EQB2	*LTF*	Lactotransferrin	4	1.22	1.45 × 10^−3^
P14618-2	*PKM*	Pyruvate kinase PKM	1	1.23	6.53 × 10^−3^
P31146	*CORO1A*	Coronin-1A	1	1.28	1.23 × 10^−2^
P04406	*GAPDH*	Glyceraldehyde-3-phosphate dehydrogenase	6	1.30	5.43 × 10^−4^
R4GN98	*S100A6*	Protein S100	2	1.37	1.18 × 10^−4^
P68431	*HIST1H3A*	Histone H3.1	1	1.45	2.92 × 10^−2^
P06732	*CKM*	Creatine kinase M-type	9	2.11	2.21 × 10^−3^

**Table 9 biomolecules-13-01471-t009:** List of most reliable plasma proteins related to the glucose metabolism pathway between sHIE+ and the other groups. sHIE+, peripheral blood from moderate-severe HIE newborns treated with TH and with a favourable outcome.

Protein ID	Gene Name	Protein Name	Number of Peptides	Log2FC	*p*-Value
P00390-2	*GSR*	Glutathione reductase, mitochondrial	1	0.61	1.79 × 10^−2^
P37837	*TALDO1*	Transaldolase	4	0.89	1.80 × 10^−2^
P06733	*ENO1*	Alpha-enolase	4	0.9	1.11 × 10^−2^
P04075	*ALDOA*	Fructose-bisphosphate aldolase A	13	0.91	8.63 × 10^−3^
P05062	*ALDOB*	Fructose-bisphosphate aldolase B	3	0.94	4.88 × 10^−2^
P14618-2	*PKM*	Pyruvate kinase PKM	1	0.95	3.09 × 10^−2^
P18669	*PGAM1*	Phosphoglycerate mutase 1	1	0.98	1.76 × 10^−2^
P09972	*ALDOC*	Fructose-bisphosphate aldolase C	1	1.01	1.97 × 10^−3^
P04406	*GAPDH*	Glyceraldehyde-3-phosphate dehydrogenase	6	1.09	2.69 × 10^−3^
P60174-1	*TPI1*	Triosephosphate isomerase	4	1.12	1.01 × 10^−2^
P00558	*PGK1*	Phosphoglycerate kinase 1	7	1.17	9.79 × 10^−4^
B4DQJ8	*PGD*	6-phosphogluconate dehydrogenase, decarboxylating	1	1.31	1.55 × 10^−3^

## Data Availability

Data supporting reported results can be made available by writing to the corresponding author p.yip@qmul.ac.uk.

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
