# Peer review of "Newborns with Favourable Outcomes after Perinatal Asphyxia Have Upregulated Glucose Metabolism-Related Proteins in Plasma"

_biomolecules, 2023, doi:10.3390/biom13101471_

Round 1
Reviewer 1 Report
This manuscript presents the results of a study seeking to describe the blood proteomic signatures in four populations: control cord blood and postnatal blood from "mild HIE", mod-sev HIE with less severe MRI findings, and mod-sev HIE with more severe MRI findings. This work bridges a gap between two important current areas of HIE research: biomarker discovery and improvement in our understanding of the pathophysiology of injury. Despite this, there are several aspects of the manuscript that deserve the authors' attention to improve the accuracy and interpretation of their findings.
This reviewer's primary concerns/comments include:
1) Two of the primary groups are never well defined in the methods: there is no clear definition of mild HIE, which is the more concerning of the two, since the definition is still controversial to this day and was much more so when the current study was performed, nearly 10 years ago. A thorough description of the criteria used (and how it included one infant with HIE stage 0) will be helpful in interpreting the findings. Additionally, how was a "normal" infant defined for the cord blood group? What inclusion/exclusion criteria were met to ensure that they were a true "normal" control?
2) What was the reasoning for using MRI as the primary study outcome? It is usually used as a surrogate for developmental outcomes, but since the authors report developmental outcomes in the results (though they never state how they were assessed in the methods - which should be added), developmental outcome would be an infinitely better study outcome than MRI findings, especially since nearly all of the MRIs in this study were performed in the time frame where they are thought to be the least accurate (8-10 days). If the authors are unable to use follow up data to stratify sHIE+ from sHIE-, then it would be this reviewer's suggestion to eliminate those two subgroups and just have sHIE, since the relationship between early MRI (2-7 days) and outcomes has indeed been shown to be strong as the authors state, but the relationship with MRI during pseudonormalization (8-10 days) and outcomes is quite weak. This may be part of why there is no clear differentiation between the HIE groups in the PCA analyses. Limiting the number of comparisons made in the study would also significantly improve the clarity of the manuscript, as trying to keep all of the different group comparisons straight is very difficult as currently written. The authors should consider which comparisons are actually relevant to whatever question they are seeking to answer, and focus on that comparison. The rest can still be included in the results, but could be separate and deserves less discussion.
3) The authors briefly describe the differences in blood draw timing in their discussion, but this is a much bigger issue than they have discussed. HIE is a triphasic injury (plus a latent phase), and many studies have shown that blood and tissue RNA and protein expression changes significantly during the different phases. By the age at blood sample listed in Table 2, at least 15% of the samples were obtained during the latent phase (<6 hours) and some were obtained at >48 hours which is approaching the tertiary phase. This wide range is likely another reason that there was not any clear differentiation between HIE groups, as some of the proteins that are upregulated in the latent phase may be downregulated in the secondary phase (or vice-versa) and therefore the very broad range of collection times would cancel out the differences. There is nothing that can be done about this significant issue at this point in the study, but it should be more thoroughly discussed in the discussion and listed as a limitation.
4) The discussion includes a fair amount of speculation, which should be limited/removed. The discussion should focus mainly on what the authors were able to show. For instance, as they state, they cannot differentiate the effects of pathophysiology severity (mild vs. mod-sev) and hypothermia, so this can be mentioned, but the claims that are made several times in the discussion (and in fact in the title of the paper) that hypothermia plays a primary role should be removed as those are purely speculation. Similarly, the authors devote much of two paragraphs in the discussion to NADPH which, although shown by others to be important in the pathophysiology of HIE, was not one of the key findings from the current study, so it is unclear why it was highlighted so thoroughly in the discussion, which should be focused on the actual findings from the study.
5) The authors claim that normal cord blood is not an adequate control because it was clearly distinct from the other populations in the PCA analyses, but this reasoning is unclear, as that would seem a good reason for why it is an adequate control. The secondary reasoning that they provide is that the source being cord blood could contribute to the difference, but then why did the authors use it, as that reasoning is not any more valid after the study than it was before the study? That seems to be as much a commentary on the adequacy of their study design as it is on whether it is a good control.
6) Take care with copying and pasting text from other sources, even when referencing it. Lines 83-87 appear to be directly copied from the source text without the use of quotation marks.
Additional more minor comments:
- As mentioned above, the mention to therapeutic hypothermia should be removed from the title and the abstract (line 36)
- The definition of favorable outcome should be described in the abstract
- Reference 1 is not an epidemiological study so is not an appropriate reference to describe the epidemiology of HIE. Please replace with a more appropriate study
- Line 46: the point is well received that there is room for improvement in the treatment of HIE, but the NNT of 9 is actually incredibly low compared to most of the common therapies in neonatal medicine, so this sentence should be restated for accuracy
- Lines 50-56: it is not clear that this paragraph adds to the study narrative. Would suggest replacing it with a paragraph discussing the current state of biomarker literature (including the gap that this study seeks to fill)
- Lines 65-67: what is the hypothesis/aim of this study. It should be stated in this paragraph
- Lines 92-95 and 99-106 are all results, not methods, so should be moved to the appropriate section
- All figures and tables that use the group abbreviations should have those abbreviations defined in that figure/table caption
- Consider combining the two statistical methods sections into one
- Why was BCA assay used for western but Bradford assay for the plasma? Please clarify in the methods
- Lines 165-166: why was there not any control (normal infant) lanes on the westerns?
- Lines 190-194: this should mention the differences in demographics between sHIE+ and sHIE-
- Lines 200-201: what were the developmental outcomes for the sHIE- group?
- Table 2 should include (%) for each of the categorical variables. The definition of sentinel event should also be included in the caption and/or methods
- Figure 2B should either have the protein names on the right removed or the figure should be expanded vertically so that they are legible
- Lines 233-234: this sentence is discussion so should be removed from the results section
- Lines 311-313: this sentence is discussion so should be removed from the results section
- Take care to minimize biased statements (i.e. those starting with words like "interestingly")
- When discussing the results in figure 5, the authors should note the low concentrations of each of the proteins and whether the statistical differences shown are physiologically relevant. Additionally, the blot images do not seem to match the graphs. It is assumed that the dark band in the sHIE+ image corresponds to the one high deviant in that group, but both of the bands are much darker in the sHIE+ image than the mHIE in 5E, but that is not reflected in the graph. Please ensure that the images are representative of the group average
- Other than the speculation in the final sentence of the first paragraph (which was mentioned above), the authors mostly just restate results in the first two paragraphs of the discussion, without adding much, if any, interpretation of the importance or relevance of their findings. Consider adding more perspective to these paragraphs.
- Most of the glucose metabolism section has no references. Consider providing appropriate references to previous literature here
- Line 490: consider providing pertinent examples of the "additional cellular functions"
- Lines 511-514: consider moving to intro to help support a paragraph on biomarkers
- Consider providing total protein values between groups to better support the claim that it is a better/more consistent value to use for normalization compared to housekeeping proteins
Thorough spell check/grammar check to catch some minor issues would benefit the manuscript (things like line 35 missing a word between "6" and "and", line 192: "need for need for", and other small typos).
Author Response
We like to thank the reviewer for providing time from their busy schedule to review our manuscript. We have provided a point-by-point rebuttal to the reviewer's comments.

Reviewer 2 Report
General Comments
This is an interesting manuscript that used TMT 10-plex labelled liquid chromatography-tandem mass spectrometry to identify proteins that could predict outcomes in infants treated with therapeutic hypothermia (TH) to attenuate the severity hypoxic ischemic encephalopathy (HIE). Although this is the first “relatively large (n=22)” study to examine proteins in the peripheral blood of infants treated TH with HIE there are some limitations of the study that need to be addressed. Although the authors were able to show that some proteins were associated with MRI changes, changes in peripheral blood after HIE must reflect changes in all organs because infants with HIE experience injury to all major organs ie kidney, liver, heart, etc.
In addition, there are major issues with the timing of the blood sampling of the groups of infants. According to table 2, the sampling range differed by 5.5 hours between the sHIE+ and sHIE- infants. Moreover, the standard deviation of the times of sampling ranges were 11.5 h in the sHIE+ and 21.3 h in the sHIE- infants. It is well known that there is an evolution of injury over hours to days after exposure to HIE. The sampling time ranges could have affected the outcomes. Furthermore, there is no mention of the timing of initiation of TH in the sHIE+ and sHIE- infants. It is well known that earlier initiation of TH treatment results in improved outcomes.
It is of great interest that the authors identified proteins involved in glucose homeostasis. As this is a very important contributor to brain injury after HIE. However, the discussion of the impact of hypo- and hyperglycemia on outcomes of HIE after TH is totally lacking. If this is a major finding of the manuscript more discussion of the topic is necessary. Please see references provided for the author’s benefit.
Specific Comments
1. Introduction: The study is purely descriptive. What was the authors’ hypothesis at the onset of the study?
2. Material and Methods:
a. When was TH initiated in the 2 groups? This could affect the outcomes and their results. What was the timing of sampling relative to the initiation of TH in the sHIE+ and sHIE- infants? Group differences could certainly affect the results.
b. Although the authors mention that the source of the proteins are not necessarily from brain as all organs are injury by hypoxia ischemia in infants with HIE, more emphasis needs to be placed on this point.
3. Results:
a. The findings of differences in glucose metabolism pathways are very interesting.
However, the discussion of its importance relevant to outcomes of HIE after TH are limited. Articles are provided below for the authors’ consideration.
b. Serial sampling of the same infant would be ideal for the study. Infants with HIE have multiple blood tests so perhaps residual blood samples could be used for this study.
4. Discussion: Although limitations are mentioned, emphasis needs to be placed on some of the above points.
5. There are some font changes in the text.
6. Lines 453 and 454 are pure speculation and probably not even enough data to support this speculation.
References:
Tam EWY, Kamino D, Shatil AS, Chau V, Moore AM, Brant R, Widjaja E. Hyperglycemia associated with acute brain injury in neonatal encephalopathy. Neuroimage Clin. 2021;32:102835. doi: 10.1016/j.nicl.2021.102835. Epub 2021 Sep 28. PMID: 34601311; PMCID: PMC8496301.
Pinchefsky EF, Hahn CD, Kamino D, Chau V, Brant R, Moore AM, Tam EWY. Hyperglycemia and Glucose Variability Are Associated with Worse Brain Function and Seizures in Neonatal Encephalopathy: A Prospective Cohort Study. J Pediatr. 2019 Jun;209:23-32. doi: 10.1016/j.jpeds.2019.02.027. Epub 2019 Apr 11. PMID: 30982528.
Salhab WA, Wyckoff MH, Laptook AR, Perlman JM. Initial hypoglycemia and neonatal brain injury in term infants with severe fetal acidemia. Pediatrics. 2004 Aug;114(2):361-6. doi: 10.1542/peds.114.2.361. PMID: 15286217.
Guellec I, Ancel PY, Beck J, Loron G, Chevallier M, Pierrat V, Kayem G, Vilotitch A, Baud O, Ego A, Debillon T. Glycemia and Neonatal Encephalopathy: Outcomes in the LyTONEPAL (Long-Term Outcome of Neonatal Hypoxic EncePhALopathy in the Era of Neuroprotective Treatment With Hypothermia) Cohort. J Pediatr. 2023 Jun;257:113350. doi: 10.1016/j.jpeds.2023.02.003. Epub 2023 Feb 23. PMID: 36828343.
Kalogeropoulou MS, Thomson L, Beardsall K. Continuous glucose monitoring during therapeutic hypothermia for hypoxic ischaemic encephalopathy: a feasibility study. Arch Dis Child Fetal Neonatal Ed. 2023 May;108(3):309-315. doi: 10.1136/archdischild-2022-324593. Epub 2022 Dec 20. PMID: 36600516.
Parmentier CEJ, de Vries LS, van der Aa NE, Eijsermans MJC, Harteman JC, Lequin MH, Swanenburg de Veye HFN, Koopman-Esseboom C, Groenendaal F. Hypoglycemia in Infants with Hypoxic-Ischemic Encephalopathy Is Associated with Additional Brain Injury and Worse Neurodevelopmental Outcome. J Pediatr. 2022 Jun;245:30-38.e1. doi: 10.1016/j.jpeds.2022.01.051. Epub 2022 Feb 2. PMID: 35120986
Author Response

(The authors gave the same response as above.)

Reviewer 3 Report
In this study the authors applied a proteomics analysis to assess a panel of proteins in the plasma of newborns with HIE. Proteomic analysis was performed in newborns with moderate-severe HIE that had initially undergone TH, and relative controls. Analysis revealed several upregulated proteins in the TH-treated moderate-severe HIE group. These proteins were associated with glucose metabolism and resulted upregulated in newborns with favourable outcomes compared to newborns with unfavourable outcomes. They conclude that the study demonstrates that TH-treated newborns with favourable outcomes have an upregulation in 6 proteins related to glucose metabolism. These proteins are further enhanced with TH.
Although the manuscript is mainly descriptive, it adds new information on changes in blood proteins after HIE and TH and it improve knowledge in this context, since previous studies used less sophisticated analytical methods (ELISA). The experimental approach is sound and the manuscript well written. The authors also addressed limitations and strength of their results.
Minor point. Page 1 Abstract. On line 35 it seems that something is missing (“upregulation in 6…..”)
Author Response

(The authors gave the same response as above.)

Round 2
Reviewer 2 Report
I have no further comments to add to my previous review.